# Deconvolution of the Genomic and Epigenomic Interaction Landscape of Triple-Negative Breast Cancer

**DOI:** 10.3390/cancers11111692

**Published:** 2019-10-31

**Authors:** Jiande Wu, Tarun Karthik Kumar Mamidi, Lu Zhang, Chindo Hicks

**Affiliations:** 1Department of Genetics, Louisiana State University Health Sciences Center, School of Medicine, 533 Bolivar Street, New Orleans, LA 70112, USA; jwu2@lsuhsc.edu; 2Graduate Biomedical Sciences, The University of Alabama at Birmingham, 1825 University Blvd, Birmingham, AL 35233, USA; tmamidi@uab.edu; 3Department of Public Health Sciences, Clemson University, 513 Edwards Hall, Clemson, SC 29634, USA; lz3@clemson.edu

**Keywords:** triple-negative breast cancer, gene, genomic, mutation, epigenomic alterations

## Abstract

Triple-negative breast cancer (TNBC) is the most aggressive form of breast cancer. Emerging evidenced suggests that both genetics and epigenetic factors play a role in the pathogenesis of TNBC. However, oncogenic interactions and cooperation between genomic and epigenomic variation have not been characterized. The objective of this study was to deconvolute the genomic and epigenomic interaction landscape in TNBC using an integrative genomics approach, which integrates information on germline, somatic, epigenomic and gene expression variation. We hypothesized that TNBC originates from a complex interplay between genomic (both germline and somatic variation) and epigenomic variation. We further hypothesized that these complex arrays of interacting genomic and epigenomic factors affect entire molecular networks and signaling pathways which, in turn, drive TNBC. We addressed these hypotheses using germline variation from genome-wide association studies and somatic, epigenomic and gene expression variation from The Cancer Genome Atlas (TCGA). The investigation revealed signatures of functionally related genes containing germline, somatic and epigenetic variations. DNA methylation had an effect on gene expression. Network and pathway analysis revealed molecule networks and signaling pathways enriched for germline, somatic and epigenomic variation, among them: Role of BRCA1 in DNA Damage Response, Hereditary Breast Cancer Signaling, Molecular Mechanisms of Cancer, Estrogen-Dependent Breast Cancer, p53, MYC Mediated Apoptosis, and PTEN Signaling pathways. The investigation revealed that integrative genomics is a powerful approach for deconvoluting the genomic-epigenomic interaction landscape in TNBC. Further studies are needed to understand the biological mechanisms underlying oncogenic interactions between genomic and epigenomic factors in TNBC.

## 1. Introduction

Triple-negative breast cancer (TNBC) is a heterogeneous disease, characterized by aggressive clinical behavior, poor prognosis, a significantly increased risk of relapse, and shorter survival rates than patients affected by other molecular subtypes of breast cancer [1,2,3]. TNBC is clinically defined as tumors that lack expression of the estrogen receptor (ER), progesterone receptor (PR), and HER2 amplification [1,2,3]. It affects primarily younger premenopausal women and tends to have higher incidences in African American women, although recent studies reported no differences in clinical outcomes between Caucasian and African American women after adjusting for socioeconomic factors [2,3,4]. Currently, there are no effective targeted therapies. Cytotoxic chemotherapy remains the only effective therapeutic modality for this aggressive and often lethal type of breast cancer [3]. Therefore, there is an urgent need for the discovery of molecular markers and targets for the development of novel targeted therapeutics. 

Our understanding of the molecular basis of TNBC and development of effective targeted therapies has been hampered by the complexity and multifactorial etiology of the disease. Genes play a major role in the pathogenesis of TNBC [4,5,6]. The majority of TNBC patients carry mutations in the highly penetrant cancer susceptibility genes BRCA1 and BRCA2 [2,3,4]. In unselected TNBC cases, the prevalence of pathogenic germline *BRCA1* and *BRCA2* mutations is approximately twice as high as in breast cancer overall [5]. Apart from *BRCA1* and *BRCA2*, the rarely mutated breast cancer predisposition genes *PALB2* and *FANCM* have been associated with TNBC [5]. Multigene panel testing has also identified genes with high and moderate penetrance associated with an increased risk of developing TNBC [6]. However, although genes may play a strong role, molecular epidemiology studies suggest that TNBC risk is largely determined by interactions between genes and environment. Increased attention has focused on epigenetic variation such as DNA methylation and TNBC because enduring epigenetic landmarks define the tumor microenvironment [7,8]. Moreover, because DNA methylation regulates gene expression, aberrant methylated genes could serve as complementary diagnostic tools, prognostic markers and predictors of response to treatment [8,9,10]. Thus, a critical knowledge gap, and an unmet medical need, is understanding the interplay between genetic (both germline and somatic) variation and epigenomic variation in TNBC.

Advances in microarray technology have enabled molecular classification of subtypes of TNBC [11,12,13]. At least, several clinically validated prognostic assays including the Prosigna PAM50 [14,15], MammaPrint [16] and Oncotype DX [16], have been developed using transcription profiling. However, although these primary analyses have been successful in identifying prognostic markers, they have been unsuccessful in establishing the causal association between gene expression and the disease. High-throughput genotyping using genome-wide association studies (GWAS) has enabled discovery of genetic variants associated with an increased risk of developing TNBC [17,18,19]. The recent surge of next generation sequencing of tumor genomes by large multicenter projects, such as The Cancer Genome Atlas (TCGA) and the International Cancer Genome Consortium (ICGC), have generated comprehensive catalogues of somatic mutations associated with TNBC [20,21]. The clonal and mutational spectrum of primary TNBCs have also been characterized [22]. We recently reported oncogenic interactions between genes containing germline and somatic mutations in TNBC [17].

However, to date, information on germline, somatic and gene expression variation has not been leveraged and integrated with DNA methylation data to map the interplay between genetic and epigenetic variation in TNBC. This limited progress must be balanced against the recognition that genomic and epigenomic alterations have long been considered as two separate molecular mechanisms participating in TNBC pathogenesis. The objective of this investigation is to deconvolute the genomic and epigenomic interaction landscape, and to discover and characterize the molecular networks and signaling pathways perturbed by these interactions in TNBC. Our working hypothesis was that TNBC originates from a complex interplay between genomic (both germline and somatic) variation and epigenomic variations. We further hypothesized that these complex arrays of interacting genomic and epigenomic factors affect entire molecular networks and signaling pathways which, in turn, drive TNBC. We addressed these hypotheses by integrating information on germline variation from GWAS with information on somatic and epigenomic variation from TCGA using gene expression as the intermediated phenotype.

## 2. Materials and Methods

The scientific premise of this investigation is that the development and progression of TNBC is complex and involves the interplay between genetic and epigenetic factors. This complexity challenges the traditional use of single-platform study design, and calls for an integrated approach to data analysis for the discovery of clinically actionable biomarkers and targets for the development of novel therapeutics. Here, we present a novel integrative analysis strategy for integrating information on germline, somatic and epigenetic variation using gene expression data as the intermediate phenotype. The overall project design and integrated analysis workflow is presented in Figure 1. Information about the original data sets used in this study is presented in Table 1. Additional information on sources of data, data processing, quality control, analysis and integration is provided in the sections below.

We used publicly available data, which has been well catalogued, annotated and linked with clinical information. The methods of data generation and technology platforms used, including experimental protocols, have been well documented by the data originators as documented below. Here, we describe the sources of data used, methods of data processing, analysis, and integration strategies used. Our analysis approach assumes that TNBC is an emergent property of genomically and epigenomically altered functionally related genes interacting in gene regulatory networks and signaling pathways. Additionally, our analysis approach uses DNA methylation as a surrogate variable representing environmental perturbations, because enduring epigenetic landmarks define the cancer microenvironment [7].

### 2.1. Germline Mutations and Associated Genes

We used germline variation information from a comprehensive catalogue which was compiled using published reports on GWAS that we have developed and published [17,23]. Details on methods used in data collection have been reported in our earlier published reports [17,23], which were based on guidelines proposed by the Human Genome Epidemiology Network for systematic review of genetic associations [24,25,26,27,28]. The information in our catalogue was supplemented by information from the GWAS Catalog, which delivers a continuously updated high-quality curated collection of all published genome-wide association studies [29]. The complete data set included SNP identification numbers (rs-IDs) linked with gene names, and their chromosome positions and evidence of association as determined by the GWAS P-value. The combined data set included 825 genes containing genetic variants associated with an increased risk of developing breast cancer, derived from GWAS reports covering a cohort population of >300,000 cases and >300,000 controls (Table 1) [17,23,29]. It is worth noting that GWAS adopted a case-control study design, and the majority (>95%) of GWAS have not been breast cancer type-specific. Accordingly, in this investigation, we followed the same design approach by considering all genetic variants and genes associated with an increased risk of developing breast cancer. A complete list of genetic variants and genes associated with an increased risk of developing breast cancer, along with original sources or published reports from which they were derived, is presented in Appendix A provided as supplementary data to this report.

### 2.2. Gene Expression and DNA Methylation Data and Somatic Mutation Information

We used gene expression and DNA methylation (array-based) data, somatic mutation and clinical information from the TCGA (https://www.cancer.gov/about-nci/organization/ccg/research/structural-genomics/tcga) [30]. We downloaded somatic mutation, gene expression and DNA methylation data, along with clinical information from TCGA, via the Genomics Data Commons (https://gdc.cancer.gov/) data portal using the data transfer tool [31]. The original data set consisted of 223 samples distributed as follows: 110 tumor samples from patients diagnosed with TNBC and 113 control samples (Table 1). Gene expression data was generated using RNA-Sequencing using the Illumina HiSeq250. The original data matrix included 60,484 probes (Table 1). DNA methylation data was generated using the Illumina HumanMethylation450 BeadChip, which has been widely used for quantifying DNA methylation and has been validated [32]. The DNA methylation data matrix included 485,578 probes (Table 1). The data was processed for analysis using the DNA methylation data processing and analysis protocols [33,34] implemented in our bioinformatics pipeline for analysis of DNA methylation data. The pipeline is optimized to take into account the convolution of biological and technical variability, and the presence of a signal bias between Infinium I and II probe design types to correct for the probe design type. This was done to eliminate bias because the amplitude of the measured methylation change depends on the underlying chemistry, consistent with Illumina data analysis protocol [32,33]. The data was normalized using quantile normalization implemented in the R Package [32,33,34,35] and was also corrected for batch effects consistent with Illumina data analysis protocol [32,33,34,35].

DNA methylation data was derived from the same patient population as gene expression data. We linked methylation samples with gene expression samples using clinical information to identify samples with both measurements. Linking gene expression data with methylation data resulted in 83 patients diagnosed with TNBC tumors and 83 controls with information on both gene expression and DNA methylation (Table 1). As the same TCGA barcode structure was used for both clinical and molecular data, we used the barcodes structure to integrate patient-based clinical data with sample-based somatic mutation data. Somatic mutation data was processed to identify the number of genes containing somatic mutations and the number of somatic mutation events per gene. From this processing step, we created a catalogue of 7659 somatic mutated genes (Table 1) used in the analysis. A comprehensive list of somatic mutated genes and number of somatic mutation events per gene is presented in Appendix A.

### 2.3. Bioinformatics Analysis of Gene Expression and DNA Methylation Data

The data collection, processing and analysis workflow are presented in Figure 1. Gene expression and DNA methylation data were processed and normalized using the Bioconductor R-package LIMMA [36]. The gene expression data matrix was filtered to remove rows with missing data, such that each row has at least ≥30% data. After data preprocessing, we generated a dataset consisting of 166 samples (83 TNBCs and 83 controls) with 28,084 probe sets which were used in the analysis. The probe IDs, gene symbols and names were matched for interpretation using the Ensemble database, a database used for gene annotation of sequencing experiments on sequencing technology platforms used by the TCGA. The expression data was quantified in Transcripts Per Kilobase Million (TPM) and was first log2 (TPM+1) transformed. For gene expression data, we performed whole transcriptome analysis comparing gene expression levels between patients diagnosed with TNBC, and matched control samples to identify all significant differentially expressed genes between tumors and control samples. We used the false discovery rate (FDR) procedure [37] to adjust the p-values for multiple hypothesis testing. Additionally, we computed the log2 Fold Change (Log2 FC) defined as the median of tumors minus median of normal for each gene. Genes were ranked based on adjusted *p*-values, FDR and LogFC. For significantly differentially expressed genes containing somatic mutations, we computed the number of somatic mutation events per gene to identify the most highly somatic mutated genes. The genes were classified as highly mutated if they produced ≥3 mutation events per gene.

Proper identification of differentially methylated sites or CpGs was central in this analysis to identify a signature of aberrantly methylated genes. Thus, for DNA methylation data, we first performed quality control by processing the data to correct for batch effects. The data matrix with 485,578 probes was filtered to remove rows with missing data, such that each row has at least ≥30% data. After data preprocessing, we generated DNA methylation data set on 166 samples (83 TNBCs and 83 controls) with 383,119 probes on the same patients as those used for gene expression data. We performed quality control and normalized the data using quantile normalization. The normalized data was then used in the analysis. We performed differential methylation analysis comparing tumor samples to control samples to discover a signature of DNA methylated genes associated with TNBC. We use the beta-values (methylation values ranging from 0.0 to 1.0) to compare tumor samples to normal samples. Discovery of significant differentially methylated genes was done consistent with Illumina protocol [32,33,34]. Genes were ranked on adjusted p-values, FDR and LogFC. We used volcano plots to compare p-values and fold change. Differentially methylated CpG sites were identified using the LIMMA package implemented in R [36]. These sites were then annotated with gene symbols using Ensemble Biomart database [38]. Using gene symbols and annotated differentially methylated sites, we computed the number of methylation sites per gene focusing on differentially methylated genes to get a quantitative assessment of DNA methylation sites per gene. The methylation sites were further classified as either hypomethylated (down) or hypermethylated (up) based on gene regulation using a TCGA visualize Starburst plot [39]. The genes were then ranked based on adjusted *p*-values (*p* < 0.05) derived from differentially methylated sites. For significantly differentially methylated genes, we computed the number of methylation or CpG sites per gene. The gene was considered highly methylated if it produced ≥3 significant methylation sites.2.4. Integration of Germline, Somatic and Epigenomic Variation

As depicted in Table 1, we used a systems level integrative analysis approach using different types of omics data each providing unique information. This is because a single omics assessment provides limited insights to understand how oncogenic interactions, and cooperation between genomic and epigenomic factors drive TNBC. We performed five (5) levels of integration. At level 1, we integrated genes transcriptionally associated with TNBC with aberrantly methylated genes associated with TNBC, to discover a unified signature of aberrantly methylated genes transcriptionally associated with TNBC. The impact of DNA methylation on gene expression was assessed by using a Starburst plot of gene expression profiles against DNA methylation profiles. Level 2 integration was performed by evaluating genes transcriptionally associated with TNBC, and aberrantly methylated genes associated with TNBC, for the presence of somatic mutations to identify a signature of genes containing somatic and epigenetic variation, and transcriptionally associated with TNBC. Level 3 involved integration by evaluating genes transcriptionally associated with TNBC, and aberrantly methylated genes associated with TNBC, for the presence of germline and somatic variation to identify a signature of genes containing all three variations and are transcriptionally associated with TNBC.

### 2.4. Network and Pathway Analysis

Levels 4 and 5 integrations were higher level integrations involving network and pathway analysis, respectively, to identify molecular networks and signaling pathways enriched for germline, somatic, epigenetic and gene expression variation. For this analysis, we used the Ingenuity Pathway Analysis (IPA) software package [40]. For these analyses, we combined three sets of genes (i) genes transcriptionally associated with TNBC and contained germline, somatic and epigenetic variation, (ii) highly somatic mutated genes containing germline variation and are transcriptionally associated with the disease, but are devoid of epigenetic variation, and (iii) highly differentially methylated genes containing germline variation and transcriptionally associated with TNBC, but do not contain somatic mutations. The rationale for using this full model was to capture both cis and trans regulatory mechanisms and to include any important pathways, which could otherwise be missed by limiting the analysis only to a set of genes containing all three alterations. We computed the probability Z-scores and the log *p*-values to assess the likelihood and reliability of correctly assigning the genes to the correct molecular networks and signaling pathways, respectively. A false discovery rate was used to correct for multiple hypothesis testing in pathway analysis. The predicted molecular networks and biological pathways were ranked based on z-scores and log P-values, respectively. Gene ontology (GO) [41] analysis, as implemented in IPA, was used to classify the genes according to the molecular functions, biological process, and cellular components in which they are involved.

## 3. Results

The genomic revolution has led to an intense focus on discovery of germline and somatic mutations associated with TNBC [17,22]. One consequence of this focus has been a reduced attention on the role of epigenetics factors in the pathogenesis of TNBC. Equipped with the tools emerging from the genomics revolution, we are now in a position to map oncogenic interactions and cooperation between genomic and epigenomic drivers of TNBC, and to identify the molecular networks and signaling pathways which they control. In order to develop a more comprehensive understanding of the biological mechanisms driving TNBC, here we used systems level integrative analyses approaches integrating multiple omics data sets at various levels to deconvolute the genomic-epigenomic interaction landscape in TNBC. In this section and the subsections that follow herein, we summarize our findings.

### 3.1. Discovery of a Signature of Aberrantly Methylated Genes Associated with TNBC

To discover and characterize a signature of aberrant DNA methylated genes associated with TNBC, we compared the methylation profiles between tumors and control samples. We hypothesized that epigenomic alterations in tumors and control samples could lead to measurable changes distinguishing TNBC patients from controls. Using a nominal *p*-value (*p* < 0.05), the analysis produced a signature of 21,196 significantly differentially methylated genes distinguishing tumor samples from control samples. The number of methylation sites per gene varied considerably ranging from 1 to 992. Among the most highly differentially methylated genes included the 88 genes with ≥100 DNA methylation sites per gene. A complete list of all significantly differentially methylated genes is presented in supplementary Appendix A.

### 3.2. Discovery of a Signature of Differentially Expressed Genes Associated with TNBC

To discover and characterize a signature of significantly differentially expressed genes associated with TNBC, we compared gene expression levels between tumor and control samples. We hypothesized that genomic alterations in tumor and control samples could lead to measurable changes distinguishing TNBC samples from controls. After correcting for multiple hypothesis testing, the analysis produced a signature of 15,404 significantly (*p* < 0.05) differentially expressed genes distinguishing tumor samples from control samples, of which 10,259 genes were highly significantly (*p* < 0.0001) differentially expressed. A complete list of all significantly differentially expressed genes is presented in Appendix A.

### 3.3. Discovery of a Signature of Aberrantly Methylated Genes Transcriptionally Associated with TNBC

An important undertaking in this investigation was the identification and characterization of a signature of differentially expressed genes which were also differentially methylated, and determining the impact of DNA methylation on gene expression profiles. We hypothesized that genes transcriptionally associated with TNBC are aberrantly methylated and that epigenomic alterations in these genes could potentially impact their expression. To address this hypothesis, we performed level 1 integrative analysis combining information on the 21,196 significantly (*p* < 0.05) differentially methylated genes with 15,404 significantly (*p* < 0.05) differentially expressed genes, as explained in the data analysis subsection.

The results showing the distribution of differentially expressed and differentially methylated genes are presented in a Venn diagram in Figure 2A. The analysis revealed a signature of 12,816 significantly differentially expressed genes which were also differentially methylated, confirming our hypothesis (Figure 2A). In addition, the analysis revealed 8380 genes which were significantly differentially methylated but were not significantly differentially expressed, and 2,588 significantly differentially expressed genes which were not significantly differentially methylated (Figure 2A).

The results showing the direction of change, and the impact of DNA methylation on gene expression for the 12,816 significantly differentially expressed genes which were also differentially methylated, are presented in a starburst plot in Figure 2B. In Figure 2B, the direction of change and the impact of DNA methylation on gene expression are highlighted by the color code, and the number of genes in each color code is shown in the key of the Figure. The y-axis shows the distribution as measured by the log2 FC for differentially expressed genes computed from RNA-seq data. The x-axis shows the distribution as measured by the log2 FC for differentially methylated genes computed from DNA methylation data. Out of the 12,816 genes evaluated, 509 genes were upregulated, 40 genes were hypomethylated and down regulated, 41 genes hyper methylated and up regulated, 324 genes were hypomethylated, 1324 genes were down regulated, 162 genes (62 genes were hypomethylated and up regulated and 100 genes hyper methylated and down regulated), and 408 genes were hyper methylated (Figure 2B). Taken together, the analysis confirmed that DNA methylation had impact on gene expression profiles.

The results showing the top 30 most highly significantly differentially methylated and highly significantly differentially expressed genes, along with the number of methylation sites per gene, are shown in Table 2. The number of CpG sites per gene varied considerably, ranging from 1 to 992. A complete list of all the 12,816 significantly differentially methylated genes transcriptionally associated with TNBC is presented in Appendix A. Overall, the analysis revealed that genes transcriptionally associated with TNBC are aberrantly methylated, and that DNA methylation has impact on gene expression. The discovery of aberrant methylated genes which were transcriptionally associated with TNBC, coupled with the observed effects of DNA methylation on gene expression, highlights the value of integrative genomics analysis for deconvolution of the complex interplay between gene expression and DNA methylation, as well as assessing the impact of aberrant DNA methylation on gene expression.

### 3.4. Discovery of a Signature of Genes Containing Both Somatic and Epigenetic Variation

Tumor development and progression is driven by acquired somatic mutations, but enduring epigenetic landmarks, such as CpG islands investigated here, define the tumor microenvironment [7]. Thus, integration of somatic with epigenetic variation has the promise of discovering tumor driver genes. To investigate whether aberrantly methylated genes transcriptionally associated with TNBC harbor somatic mutations, we performed level 2 integration as explained in the methods section. We hypothesized that aberrant methylated genes transcriptionally associated with TNBC harbor somatic mutations. First, we evaluated the 7659 somatic mutated genes for their association with TNBC and aberrant DNA methylation. The results of this investigation are presented in a three-way Venn diagram representing somatic, epigenetic and gene expression variation, as shown in Figure 3.

The analysis revealed a signature of 4922 genes containing somatic, epigenomic and gene expression variation associated with TNBC, confirming our hypothesis. In addition, the analysis produced 7894 aberrantly methylated genes transcriptionally associated with the disease, 2234 aberrantly methylated genes containing somatic alterations, and 82 aberrantly methylated genes transcriptionally associated with the disease. Furthermore, the analysis revealed 421 genes containing somatic mutations only, 6146 genes containing epigenomic alterations only, and 2506 differentially expressed genes altered in the transcriptome only.

The results showing a signature of the top 30 somatic mutated genes containing epigenomic alterations and transcriptionally associated with TNBC are presented in Table 3. For the somatic mutated genes, the number of somatic mutation events per gene varied markedly, ranging from 1 to 27. Likewise, there was significant variation in the number of methylation sites per gene. Interestingly, among the genes containing epigenetic and somatic alterations significantly associated with TNBC, included the genes *TTN, MUC4, SYNEI, PIK3CA, ASPM, ARID1B* and *AHNAK,* recently reported to be potential prognostic markers for TNBC [42,43,44,45,46,47,48,49]. A complete list of all the significantly differentially expressed and aberrantly methylated somatic mutated genes are presented in Appendix A.

Overall, the analysis under level 2 integration revealed that the majority of the somatic mutated genes tend to be DNA methylated and transcriptionally associated with TNBC, confirming our hypothesis (Figure 3). The discovery of differentially expressed aberrantly methylated somatic mutated genes associated with TNBC demonstrates that integrative analysis is a powerful tool for the discovery of potential clinically actionable molecular markers, and that somatic mutations and DNA methylation are likely to cooperate in driving TNBC.

### 3.5. Discovery of a Signature of Genes Containing Germline, Somatic and Epigenomic Variation

Our primary hypothesis in this investigation was that TNBC is a complex disease influenced by both inherited variants in the germline DNA, somatic mutations acquired during tumorigenesis, and epigenomic alterations. To address this hypothesis, we performed level 3 integration, integrating information on germline, somatic, epigenomic, and gene expression variation to discover a unified signature containing all the four variations. As first step in this analysis, we evaluated the 825 genes containing germline mutations for association with TNBC using transcriptome and methylation data, and for the presence of somatic mutations. The scientific premise for performing this evaluation was to infer the causal association between gene expression and DNA methylation with the disease. This analysis step was necessary because, as noted earlier in this report, GWAS has not been breast cancer type-specific or subtype-specific.

The results of this integrative analysis are shown in a four-way Venn diagram in Figure 4. The analysis produced a signature of 228 genes containing somatic, germline and epigenomic alterations transcriptionally associated with TNBC, confirming our hypothesis. In addition, the analysis produced 259 genes containing germline and epigenomic variation significantly associated with the disease, 67 genes containing germline and epigenetic variation, and 19 genes containing germline mutations only associated with TNBC. Furthermore, the analysis produced a signature of 152 genes containing germline mutations only, 414 genes containing somatic mutation and epigenomic alterations only, 6056 genes with epigenetic alterations only, and 2487 genes perturbed in the transcriptome but without somatic, germline or epigenomic alterations.

The results showing the top 30 genes containing somatic, germline and epigenomic variation transcriptionally associated with TNBC are presented in Table 4. Also presented in Table 4 are the number of somatic mutations and methylation sites per gene, along with association, expression and methylation *p*-values. A complete list of genes containing somatic, germline and epigenomic variation transcriptionally associated with TNBC is presented in Appendix A, provided as supplementary data to this report.

As can be seen from Table 4 and accompanying Appendix A, the distribution of germline, somatic and epigenetic variation varied markedly per gene. Among the genes containing germline, somatic and epigenomic variation transcriptionally associated with TNBC, included the genes *BCRA1, BRCA2, PTEN* and *TP53* with high penetrance, and the genes *CHEK2, BRIP1, RAD51, CDKN2A, BARD1, MSH2, ATM* and *PALB2,* with moderate penetrance [6,50]. A subset of these genes including *BRCA1, BRCA2, TP53, MSH3, ATM,* and *MSH6* are involved in DNA repair [6,50]. This is a significant finding given that epigenomic alterations in DNA repair genes leads to a compromise in the genome integrity and ultimately to carcinogenesis [45,51].

As noted earlier in this report, one of the challenges of using genetic susceptibility variants and associated genes is that breast cancer consists of two different types (TNBC and non-TNBC) and many molecular subtypes. Compounding this problem is that early GWAS on breast cancer were not breast cancer type-specific or subtype-specific. To address this knowledge gap, and to determine whether any of the genes containing germline, somatic and epigenetic variation transcriptionally associated with TNBC discovered in this investigation have been directly associated withTNBC susceptibility specifically, we performed *in silico* validation by evaluating the genes containing all three variations against the literature in our catalogue, and provided herein as Appendix A.

The evaluation revealed that the genes *BRCA1, BRCA2, TP53, ATM, CHEK, PALB2, FANCM, RAD50, BARD1, PTEN, XRCC2, STK11, BRIP1, LGR6, TERT, ESR1, TOX3, PEX14, ADAM29, EBF1, TCF7L2, PTHLH, NTN4, RAD51L1, RAD51D, RAD51C, MLK1, MDM4, FTO, MAP3K1, LSP1, TGFB1, CASP8, TGF10, CDKN2B, CDKN2A, ANKRD16, FBXO18, ZNF365, ZMIZ1, FGFR2, LSP1, MYEOV, COX11* and *ANKLE1* have been directly associated with TNBC susceptibility [17,52,53,54,55], validating our findings.

### 3.6. Molecular Networks and Signaling Pathways Enriched for Germline, Somatic and Epigenomic Variation

To comprehensively investigate the potential oncogenic interactions and cooperation between the genetic (both germline and somatic) variation and epigenomic variation, we performed higher level integration using information on germline, somatic, epigenomic and gene expression variation. Level 4 integration involved network analysis and level 5 integration involved pathway analysis, as explained in the methods subsection. We hypothesized that TNBC originates from a complex interplay between genomic (both germline and somatic mutations) variation and epigenomic variation, and that these complex arrays of interacting genomic and epigenomic factors affect entire molecular networks and biological pathways which, in turn, drive TNBC. The rationale was that the biological mechanisms of action driving TNBC happen both in cis and in trans. Thus, we sought to discover functionally related genes interacting in gene regulatory networks and signaling pathways.

The results of network analysis are presented in Figure 5. In this Figure, the genes containing germline, somatic and epigenomic alterations are in red fonts, highly somatic mutated genes containing epigenomic changes are shown in blue fonts, and highly epigenetically altered genes containing germline mutations are shown in green fonts. Network analysis produced 25 gene regulatory networks with Z-scores ranging from 13 to 54. The discovered networks contained functionally related genes with overlapping functions. Network analysis revealed genes predicted to be involved in cancer, organismal injury and abnormalities, cellular assembly and organization, cell-to-cell signaling and interaction, cellular assembly and organization, cell cycle, cellular development, inflammatory disease, inflammatory response, DNA replication, recombination and repair, cell death and survival, immunological disease, cellular function and maintenance, cellular development, cellular growth and proliferation, cellular movement, post-translational modification, and protein synthesis.

Among the genes revealed by network analysis included the genes *BRCA1, BRCA2, PALB2, FANCG, FANCA, CREB5, ELL, SATB1, DOT1L, FOXM1, ATXN1, PTPN7, INCENP, EGFR, NUMA1, PTPN22, FAF1, EHMT2, GREB1, ZBTB38, XRCC6, RNF146, TCF7L2, MYO10, CEP55, PTPN23, NF2* and *POM1* containing germline, somatic and epigenetic variation, confirming our hypothesis that interactions among these genes affect entire molecular networks (Figure 5). Interestingly, network analysis produced the genes *CHEK1, CHEK2, XRCC2, XRCC3, FANCE, POLB, BARD1, CDKN2A, CDKN1A, XPC, RBX1, ESR1, BCAS3, XRCC4, IST1, XRCC1, PGR, LMO4, FYN, IGF1R, CCND1, TNF, ERCC2, NELFA, CDK6, EWSR1, POLR2J, AMFR* and *UBE2T* containing germline and epigenetic variation, interacting with genes containing somatic and epigenomic variation (Figure 5). The analysis also produced the genes *BAG6, CTBP1, PFKP, IGF2, JARID2, IGHG3, RCN1, HOOK2* and *SPTBN1* containing somatic mutations and epigenomic alterations (Figure 5). This demonstrates that oncogenic interactions and cooperation between genetic and epigenomic alterations are likely to occur partially through gene regulatory networks.

Interestingly, network analysis revealed the genes *BRCA1, BRCA2, PTEN* and *TP53* which have high penetrance, and the genes *CHEK2, BRIP1, RAD51, CDKN2A, BARD1, MSH2, ATM* and *PALB2* with moderate penetrance [6,49], are functionally related and interact with other genes of unknown level of penetrance (Figure 5). This suggests that genes with high and moderate penetrance may be regulating their downstream target genes with low penetrance and genes with unknown penetrance. Crucially, the results of network analysis revealed that both the genes containing germline mutations strongly, and weakly associated with breast cancer, interact with somatic mutated and aberrantly methylated genes (Figure 5). The results of network analysis confirmed interactions and cooperation among the genes containing germline, somatic and epigenomic variation.

To further gain insights and understand the broader biological context in which germline, somatic, epigenomic and gene expression variation operate, and to establish putative functional bridges between genomic and epigenomic interactions and the pathways they regulate, we performed pathway analysis. Our working hypothesis here was that oncogenic interactions and cooperation among genes containing germline, somatic and epigenetic variation affect signaling pathways. We sought to discover signaling pathways enriched for germline, somatic and epigenetic variation.

The results of pathway analysis are presented in Figure 6. The analysis produced multiple signaling pathways enriched for germline, somatic and epigenetic variation (Figure 6). Among the discovered signaling pathways included those implicated in TNBC including, Role of *BRCA1* in DNA Damage Response, Hereditary Breast Cancer Signaling, DNA Double-Strand Break Repair by Non-Homologous End Joining, ATM Signaling, Molecular Mechanisms of Cancer, Estrogen-Dependent Breast Cancer Signaling, DNA Double-Strand Break Repair by Homologous Recombination, Cell Cycle: G1/S Checkpoint Regulation, p53 Signaling, FGF Signaling, Role of CHK Proteins in Cell Cycle Checkpoint Control, Estrogen-mediated S-phase Entry, IL-7 Signaling Pathway, MYC Mediated Apoptosis Signaling, HER-2 Signaling in Breast Cancer, ILK Signaling, NF-kB Signaling, PTEN Signaling, and Mismatch Repair in Eukaryotes (Figure 6).

The results of pathway analysis confirmed our working hypothesis that TNBC originates from a complex interplay between genetic and epigenetic variation, and that these complex array of interacting genomic and epigenomic factors are likely to affect entire signaling pathways likely to drive the disease. The discovery of multiple signaling pathways enriched for germline, somatic and epigenomic variation, suggests that pathway crosstalk may be involved in the development and progression of TNBC. Taken together, the results of network and pathway analysis demonstrate that in the context of TNBC, the disease state can be considered an emergent property of molecular networks and signaling pathways regulated by genetic, somatic and epigenomic variation. In summary, the results show that integrating large-scale, high-dimensional genomic and epigenomic data using transcriptome data as the intermediate phenotype holds the promise of defining the molecular networks and signaling pathways that directly respond to genomic and environmental perturbations associated with TNBC, and are likely to drive the disease.

## 4. Discussion

We performed a systems level integration of information on germline, somatic, epigenomic and gene expression variation to deconvolute the genomic and epigenomic interaction landscape in TNBC. We discovered signatures of functionally related genes enriched for germline, somatic and epigenetic variation transcriptionally associated with TNBC. More importantly, we discovered molecular networks and signaling pathways enriched for germline, somatic and epigenetic variation, suggesting that oncogenic interactions and cooperation between genomic and epigenomic variation is likely to affect entire network states and signaling pathways likely to drive TNBC. Several studies have performed integrative analysis of omics data in overall breast cancer using TCGA data [56,57,58,59]. Zhang et al. reported identification of novel prognostic indicators for TNBC patients through integrative analysis of cancer genomics and protein interactome data [56]. Berger et al. reported a comprehensive pan-cancer molecular study of gynecological and breast cancers [57], and Shilpi et al. reported identification of genetic and epigenetic variants associated with breast cancer prognosis by integrative analysis [58]. Aberrant methylation in breast cancer has also been reported [59,60].

The main differences between these earlier studies and our study include: (1) our study focuses on TNBC, the most aggressive and lethal form of breast cancer, whereas reported studies [56,57,58,59,60] focused on breast cancer in general. (2) Our study integrates germline, somatic, epigenetic and gene expression variation, and assesses the potential impact of DNA methylation on expression of genes altered in the germline and in the tumor genomes, a phenomenon not previously reported. (3) Most notably, the integration of germline mutation information from GWAS, and somatic and epigenomic variation from TCGA to infer the causal association between gene expression and TNBC, has not been reported. To our knowledge, this is the first investigation to undertake this comprehensive approach to map the germline, somatic, epigenomic and gene expression variation interaction landscape in TNBC. Here, we discuss the significance of these findings in the context of translational and potential clinical applications to guide patient care and therapeutic decision making.

### 4.1. Integrating Transcription with DNA Methylation Profiling

The discovery of a signature of aberrantly methylated genes transcriptionally associated with TNBC suggests that DNA methylation markers could be used to complement gene expression based prognostic markers. For example, hypermethylated genes discovered in this investigation could be used as biomarkers because hypermethylation of the CpG islands in the promoter regions of tumor suppressor genes is a major event in the origin of many cancers including TNBC [8]. Hypermethylation of the CpG islands promoter can affect genes involved in cell cycle, DNA repair, cell-to-cell signaling and apoptosis, many of which were discovered in this study [8]. Most notably, although we did not test the prognostic value and the ability of aberrantly methylated genes discovered in this investigation to function as drug targets, several studies have identified aberrant methylated genes as potential drug targets [8,9,10]. For example, the ability of DNA methylation to predict response of TNBC to all-trans retinoic acid has been reported [9]. DNA methyltransferase expression as predictor of sensitivity to decitabine [10], and epigenetic silencing of TNBC hallmarks by Withaferin A have been reported [61]. Thus, such markers, if confirmed, could be used to complement molecular markers derived from transcription profiling.

### 4.2. Integrating Somatic Variation with Epigenomic Variation

Tumor development and progression is driven by acquired somatic driver mutations. However, enduring epigenetic landmarks define the cancer microenvironment [7]. Crucially, epigenetic perturbations are involved in gene silencing and allow cancer cells to adapt to changes in their microenvironment [8]. Thus, aberrantly methylated oncogenes such as hypermethylated genes discovered in this study, if confirmed, could represent important molecular biomarkers to guide therapeutic decision making. Although we did not evaluate the prognostic values of the discovered genes in this investigation, several studies have reported the prognostic value of the genes such as *BRCA1, BRCA2,* and *PTEN* discovered in this study, using transcription profiling [14,15,16], and DNA methylation profiling [8,9,10]. The novel aspect of our investigation is that it proposes combining transcription with DNA methylation profiling for the discovery of potential clinically actionable molecular markers.

### 4.3. Oncogenic Interactions between Genes Containing Germline and Epigenetic Variation

We discovered genetically altered and aberrantly methylated TNBC predisposition genes including *BRCA1, BRCA2, ATM, PALB2, CDH1, TP53, FANCM, CHEK2, MLH1, MSH2, MSH6* and *PMS2.* This is consistent with the previous report on DNA methylation [62]. The novel aspect of our approach is that, by integrating information on epigenetic variation with genetic variation, it has the promise to explain the missing variation not explained by genetic variants derived from GWAS reported thus far. Such information may also be amenable for inclusion in risk prediction models, such as polygenic risk scores, to identify patients at high risk of developing TNBC [63,64,65]. Indeed, one caveat is important here. We did not investigate allele-specific DNA methylation or allele-specific expression to determine whether they would serve as methylation quantitative trait loci (mQTLs) or expression quantitative trait loci (eQTLs) [58]. This was partially because germline mutation information from GWAS is derived from diverse populations for which there was neither gene expression nor DNA methylation. Despite this limitation, our investigation revealed that aberrant DNA methylation affects gene expression. Moreover, allele-specific methylation has been shown to be prevalent and to be contributed by CpG-SNPs interactions in the human genome [66], and differential allele-specific expression has been shown to uncover breast cancer genes dysregulated by cis noncoding mutations [67].

### 4.4. Disease Networks and Pathways as Potential Therapeutic Targets

The discovery of multiple signaling pathways enriched for germline, somatic and epigenetic variation is of particular interest. These signaling pathways such as Role of *BRCA1* in DNA Damage Response, Hereditary Breast Cancer, ATM, Molecular Mechanisms of Cancer, Estrogen-Dependent Breast Cancer, Cell Cycle Regulation, p53 Signaling, FGF Signaling, Role of CHK Proteins in Cell Cycle Checkpoint Control, MYC Mediated Apoptosis, NF-kB, and PTEN Signaling pathways could serve as targets for the development of novel therapeutics. Given the lack of effective targeted therapies for TNBC currently, development of novel therapeutics could significantly improve care, and potentially reduce the mortality rate attributable to TNBC. For example, a novel pathogenic BRCA1 germline mutation promoting TNBC cell progression, and enhancing sensitivity to DNA damage agents was recently discovered [68]. Patients carrying this mutation may benefit from DNA damaging treatment regimens [68]. CHK proteins may also be used for selective targeting the vulnerability of RB tumor suppressor loss in TNBC [69]. Recently, it was shown that the mutant p53 has the promise as a therapeutic target for the treatment of TNBC in preclinical investigation with the anti-p53 drug, PK11007 [70]. Thus, taken together and if confirmed, these pathways have the promise to serve as potential therapeutic targets.

### 4.5. Limitations and Future Research Directions

We are aware and mindful of the limitations of using germline mutation information from GWAS reports, somatic mutations, epigenomic and gene expression variation from TCGA. Both GWAS and TCGA projects are heavily biased towards women of European ancestry. Studies including other ethnic populations, such as African American women who are disproportionately impacted by TNBC, may yield additional useful information and are highly recommended, not only for providing scientific knowledge but also for the realization of precision medicine, and precision prevention without exacerbating health disparities in TNBC. This is particularly important in light of a recent study reporting differences in the mutation landscape of TNBC in African American women and Caucasian women [71]. Our investigation was driven by the use of available data resources from GWAS and TCGA as a cost effective way of addressing and gaining insights on a longstanding problem of oncogenic interactions, and cooperation between genomic and epigenomic variation and their potential joint role in TNBC. Another limitation is that integration of GWAS information with TCGA here focused on establishing the causal association between gene expression and TNBC, but makes it challenging to link the information to clinical outcomes and to validate the findings in an independent data set. This limitation is inherent in integration of disparate data sets, which is beyond the scope of the investigation. Despite these limitations, which we readily acknowledge, the study provided insights about possible oncogenic interactions between genomic and epigenomics alterations and the broader biological context in which they operate.

## 5. Conclusions

We discovered a signature of aberrantly methylated genes transcriptionally associated with TNBC and showed that aberrant DNA methylation affects the expression of genes involved in TNBC. We discovered molecular networks and signaling pathways enriched for germline, somatic, epigenomic and gene expression variation. We conclude that integrative analysis is a powerful approach to deconvoluting the oncogenic interactions between genomic and epigenomic variation, and for the discovery of potential driver genes in TNBC. Further research is recommended to understand the biological mechanisms underlying oncogenic interactions and cooperation between genomic and epigenomic factors, and to link genomic-epigenomic interactions to clinical outcomes.

## Figures and Tables

**Figure 1 cancers-11-01692-f001:**
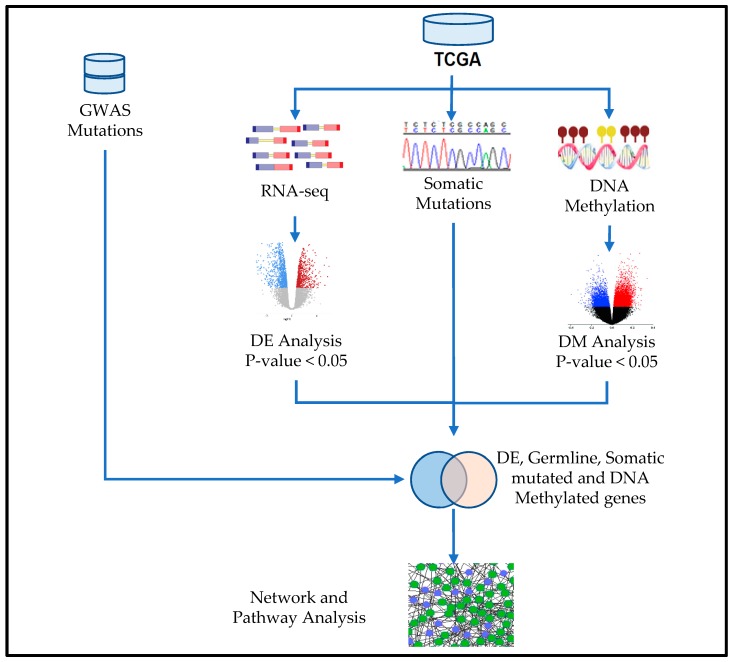
Project design and integrated genomic analysis workflow integrating information on germline, somatic and epigenomic variations using gene expression data as the intermediate phenotype.

**Figure 2 cancers-11-01692-f002:**
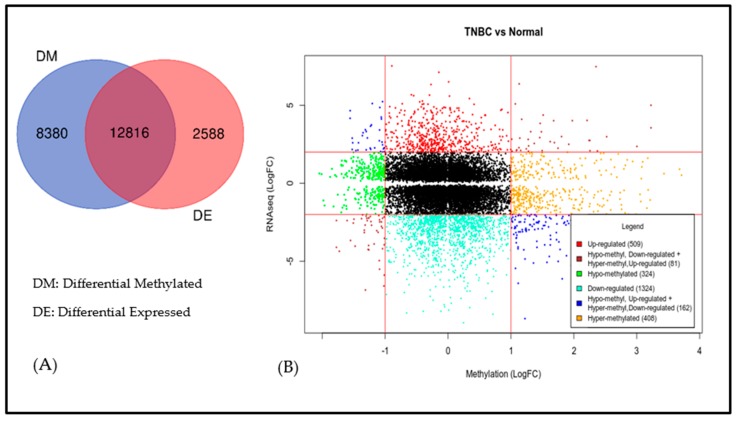
Signature of aberrantly methylated and differentially expressed genes transcriptionally associated with triple-negative breast cancer (TNBC). (**A**) Venn diagram representing genes significantly differentially expressed and significantly differentially methylated between tumors and controls. The intersection shows genes containing both genomic and epigenomic alterations and are associated with TNBC. (**B**) Two-way Starburst plot showing differentially expressed genes (y-axis) and differentially methylated genes (x-axis). The colors represent direction of change and activity as represented in the Figure legend.

**Figure 3 cancers-11-01692-f003:**
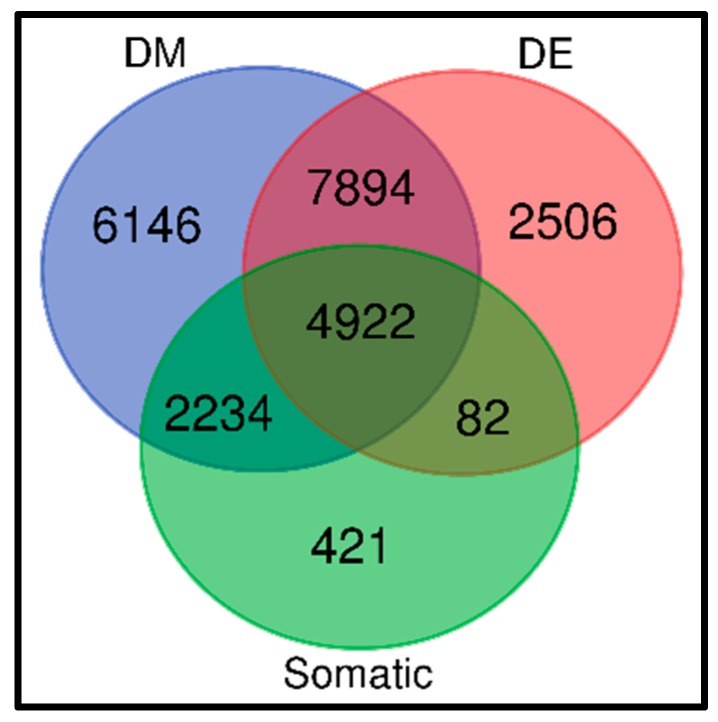
Three-way Venn diagram showing the number of transcriptionally, somatically and epigenomically altered genes associated with TNBC. DM indicates DNA methylation, DE denotes differentially expressed, and Somatic indicates somatic mutated genes.

**Figure 4 cancers-11-01692-f004:**
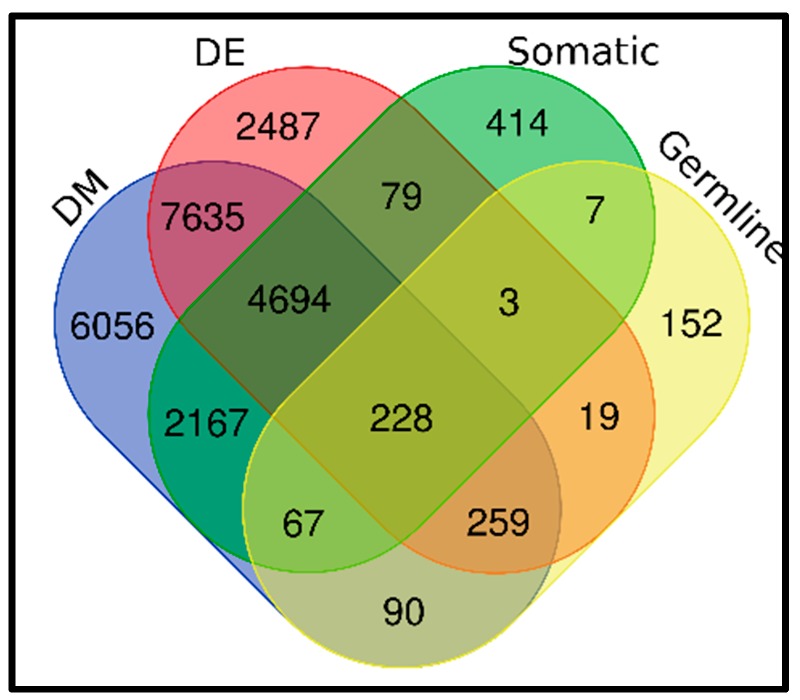
Four-way Venn diagram showing the distribution and the numbers of genomically, germline, somatically and epigenomically altered genes. DM indicates DNA methylation, DE denotes differentially expressed, and Somatic indicates somatic mutated genes.

**Figure 5 cancers-11-01692-f005:**
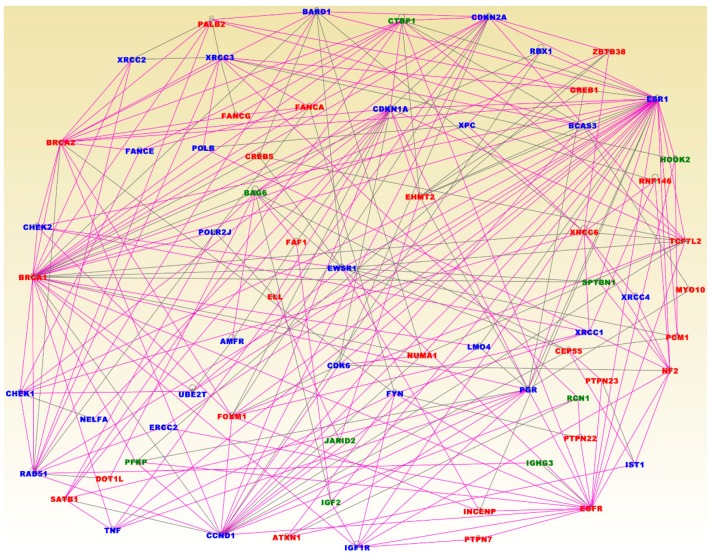
Gene regulatory networks of functionally related genes enriched for germline, somatic and epigenomic alterations. Genes containing germline, somatic and epigenomic alterations are in red fonts. Highly somatic mutated genes containing epinomic changes are shown in blue fonts. Highly epigenetically altered genes containing germline mutations are shown in green fonts.

**Figure 6 cancers-11-01692-f006:**
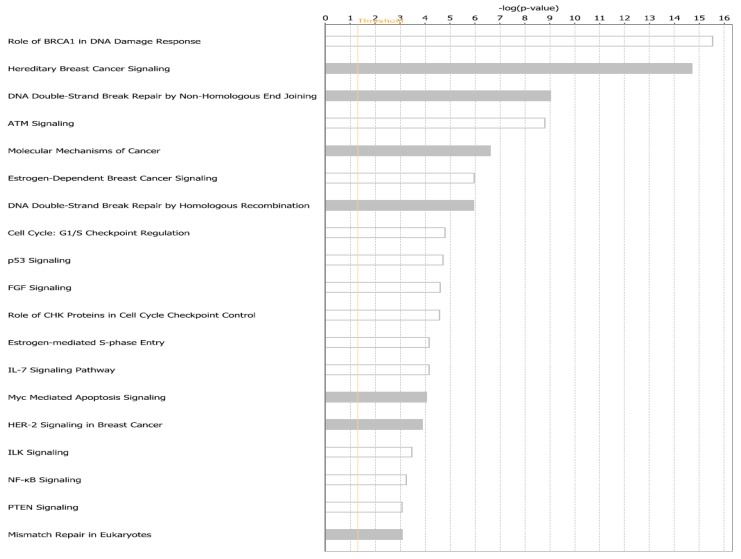
Top most significant pathways enriched for germline, somatic, and epigenomic variations for genes associated with TNBC.

**Table 1 cancers-11-01692-t001:** Characteristics and distribution of the original data sets used in the study.

Data Type	TNBC		
	Genes or Probes	Tumor Samples	Control Samples
Gene expression	60,484 Probes	110	113
Methylation	485,578 probes	83	83
Somatic	7659 genes	110	113
GWAS *	825 genes	>300,000	>300,000

* Sample estimates based on genome-wide association studies (GWAS) catalogue and GWAS reports, from which the germline and somatic mutation information was derived, and represent all breast cancers. Note that the original GWAS Reports are provided in Appendix A.

**Table 2 cancers-11-01692-t002:** A list of the top 30 highly significant differentially methylated genes which were also highly significantly differentially expressed between TNBC and controls.

Gene_Symbol	Chromosome	Methylation		RNAseq
		Event	Adjust *p*-Value	Adjust *p*-Value
RP1	8q11.23	992	3.96 × 10^−24^	8.94 × 10^−5^
PTPRN2	7q36.3	917	7.85 × 10^−24^	4.24 × 10^−16^
PRDM16	1p36.32	434	1.78 × 10^−22^	3.42 × 10^−17^
TNXB	6p21.33	364	3.11 × 10^−22^	1.29 × 10^−24^
MAD1L1	7p22.3	317	3.89 × 10^−23^	2.29 × 10^−7^
DIP2C	10p15.3	298	4.71 × 10^−23^	1.24 × 10^−9^
PCDHGA2	5q31.3	291	3.57 × 10^−24^	1.30 × 10^−19^
SNHG14	15q11.2	283	1.56 × 10^−20^	2.15 × 10^−13^
PCDHGA3	5q31.3	277	3.57 × 10^−24^	2.05 × 10^−21^
ERICH1	8p23.3	270	3.25 × 10^−23^	1.47 × 10^−2^
ADARB2	10p15.3	257	5.96 × 10^−24^	6.18 × 10^−10^
PCDHGA4	5q31	253	3.57 × 10^−24^	3.25 × 10^−15^
PCDHGB2	5q31	239	3.57 × 10^−24^	4.51 × 10^−8^
PCDHGA5	5q31	229	3.57 × 10^−24^	6.38 × 10^−15^
EIF2B5	3q27.1	217	2.57 × 10^−23^	2.26 × 10^−4^
PCDHGB3	5q31	213	3.57 × 10^−24^	2.64 × 10^−8^
TBCD	17q25.3	204	1.30 × 10^−23^	8.35 × 10^−7^
HDAC4	2q37.3	202	6.13 × 10^−24^	3.71 × 10^−8^
MCF2L	13q34	202	6.15 × 10^−24^	9.10 × 10^−5^
PCDHGA6	5q31	202	3.57 × 10^−24^	3.20 × 10^−14^
SDK1	7p22.2	202	4.35 × 10^−22^	1.13 × 10^−8^
INPP5A	10q26.3	190	3.55 × 10^−22^	4.39 × 10^−13^
PCDHGA7	5q31	188	3.57 × 10^−24^	1.69 × 10^−13^
ATP11A	13q34	177	6.26 × 10^−23^	2.83 × 10^−2^
PCDHGB4	5q31	175	3.57 × 10^−24^	6.45 × 10^−10^
KCNQ1	11p15.5	174	2.29 × 10^−22^	2.51 × 10^−3^
HOXA3	7p15.2	168	1.26 × 10^−23^	3.43 × 10^−11^
PCDHGA8	5q31.3	166	3.57 × 10^−24^	3.74 × 10^−3^
C7orf50	7p22.3	163	4.59 × 10^−23^	1.54 × 10^−7^
AGAP1	2q37.2	160	5.28 × 10^−21^	2.25 × 10^−3^

**Table 3 cancers-11-01692-t003:** List of the top 30 genes containing both somatic and epigenetic alterations transcriptionally associated with TNBC. DM represents differentially methylated.

Gene	Chromosome	Methylation		RNAseq	Somatic Events
		DM_Sites	Adjust *p*-Value	Adjust *p*-Value	
TTN *	2q31.2	20	1.53 × 10^−13^	8.34 × 10^−10^	27
MUC4 *	3q29	28	3.91 × 10^−19^	7.60 × 10^−4^	13
FAT3	11q14.3	23	9.50 × 10^−15^	1.67 × 10^−9^	12
USH2A	1q41	12	1.53 × 10^−11^	1.73 × 10^−6^	12
SYNE1 *	6q25.2	29	1.04 × 10^−18^	1.66 × 10^−20^	9
FCGBP	19q13.2	14	1.60 × 10^−17^	3.48 × 10^−2^	9
SPTA1	1q23.1	3	2.00 × 10^−14^	8.63 × 10^−5^	9
DNAH17	17q25.3	76	3.52 × 10^−21^	3.74 × 10^−4^	8
DST	6p12.1	41	1.76 × 10^−18^	1.14 × 10^−21^	8
MUC5B	11p15.5	40	9.44 × 10^−16^	1.71 × 10^−8^	8
PIK3CA *	3q26.32	7	1.11 × 10^−16^	3.26 × 10^−6^	8
PLEC	8q24.3	103	1.36 × 10^−21^	1.28 × 10^−4^	7
CSMD2	1p35.1	47	1.81 × 10^−19^	8.84 × 10^−12^	7
CREBBP	16p13.3	38	1.40 × 10^−21^	1.85 × 10^−3^	7
FLG	1q21.3	33	3.22 × 10^−17^	4.48 × 10^−9^	7
KMT2D	12q13.12	17	3.06 × 10^−17^	6.93 × 10^−4^	7
AHCTF1	1q44	10	1.20 × 10^−7^	3.48 × 10^−4^	7
ASPM *	1q31.3	10	1.17 × 10^−12^	5.39 × 10^−24^	7
MYO18B	22q12.1	4	4.93 × 10^−16^	3.73 × 10^−6^	7
USP34	2p15	4	2.38 × 10^−3^	1.36 × 10^−2^	7
KIF26B	1q44	79	8.82 × 10^−15^	7.07 × 10^−16^	6
SPTBN1	2p16.2	54	4.56 × 10^−24^	8.06 × 10^−21^	6
LRP1	12q13.3	52	5.94 × 10^−23^	1.08 × 10^−17^	6
COL18A1	21q22.3	47	1.17 × 10^−21^	1.41 × 10^−3^	6
ARID1B *	6q25.3	43	1.71 × 10^−16^	5.95 × 10^−3^	6
ZNF512B	20q13.33	42	3.57 × 10^−24^	4.92 × 10^−5^	6
AHNAK *	11q12.3	30	8.81 × 10^−23^	1.94 × 10^−24^	6
CACNA1B	9q34.3	24	6.13 × 10^−15^	1.31 × 10^−7^	6
STAB1	3p21.1	19	1.17 × 10^−17^	2.65 × 10^−4^	6
LAMA3	18q11.2	18	1.13 × 10^−22^	7.72 × 10^−19^	6

* Genes reported as potential prognostic markers for TNBC [42,43,44,45,46,47,48,49].

**Table 4 cancers-11-01692-t004:** **Top list of genes in the d**iscovered signature of genes containing somatic, germline and epigenetic vaariation. DM represents differentially methylated.

Gene Symbol	Chromosome	Methylation		RNAseq	Somatic Events	GWAS		
		DM Sites	Adjust *p*-Value	Adjust *p*-Value		SNP	*p*-Value	Event
*BRCA1*	17q21.31	12	2.01 × 10^−5^	2.78 × 10^−3^	5	rs1799950	2.00 × 10^−4^	2
*FHOD3*	18q12.2	11	8.83 × 10^−14^	9.05 × 10^−11^	5	rs9956546	2.90 × 10^−6^	2
*MYO10*	5p15.1	45	1.86 × 10^−19^	2.29 × 10^−11^	4	rs2562343	9.20 × 10^−3^	2
*CNTNAP2*	7q35	30	8.36 × 10^−17^	1.83 × 10^−3^	4	rs10487920	3.90 × 10^−4^	2
*RELN*	7q22.1	12	1.92 × 10^−13^	2.94 × 10^−18^	4	rs17157903	1.00 × 10^−2^	2
*MSH3*	5q14.1	10	2.46 × 10^−10^	7.38 × 10^−15^	4	rs6151904	1.24 × 10^−2^	2
*ATM*	11q22.3	20	9.88 × 10^−7^	1.47 × 10^−5^	3	rs1801516	2.00 × 10^−4^	2
*MTHFR*	1p36.22	9	7.33 × 10^−12^	1.33 × 10^−6^	3	rs180113	4.10 × 10^−2^	2
*PALB2*	16p12.2	3	1.54 × 10^−2^	1.87 × 10^−3^	3	deletion	4.00 × 10^−4^	2
*FBXL7*	5p15.1	35	2.85 × 10^−19^	9.65 × 10^−12^	2	rs12652447	5.60 × 10^−4^	2
*NUMA1*	11q13.4	28	7.97 × 10^−20^	1.17 × 10^−7^	2	rs3750913	1.00 × 10^−2^	2
*RB1*	13q14.2	24	2.24 × 10^−21^	9.05 × 10^−11^	2	rs2854344	7.00 × 10^−3^	2
*AACS*	12q24.31	19	1.04 × 10^−15^	1.12 × 10^−3^	2	rs7307700	2.00 × 10^−2^	2
*WRN*	8p12	15	3.30 × 10^−16^	3.19 × 10^−4^	2	rs1346044	2.00 × 10^−2^	2
*GRIN3A*	9q31.1	12	7.03 × 10^−11^	8.98 × 10^−5^	2	rs10512287	2.30 × 10^−4^	2
*BID*	22q11.21	11	3.11 × 10^−22^	3.39 × 10^−10^	2	rs8190315	1.00 × 10^−2^	2
*DMBT1*	10q26.13	11	3.79 × 10^−18^	1.85 × 10^−7^	2	rs11523871	2.00 × 10^−3^	2
*FOXM1*	12p13.33	8	1.67 × 10^−6^	2.57 × 10^−24^	2	rs2074985	3.40 × 10^−2^	2
*MSH6*	2p16.3	8	1.11 × 10^−11^	5.51 × 10^−10^	2	rs3136337	3.39 × 10^−2^	2
*MTR*	1q43	4	7.78 × 10^−10^	7.42 × 10^−3^	2	rs1805087	2.00 × 10^−2^	2
*DSEL*	18q22.1	2	2.37 × 10^−14^	4.19 × 10^−19^	2	rs17827708	9.00 × 10^−3^	2
*FANCG*	9p13.3	2	1.85 × 10^−16^	8.14 × 10^−14^	2	rs4986940	2.79 × 10^−2^	2
*EHMT2*	6p21.33	99	6.80 × 10^−20^	4.19 × 10^−12^	1	rs535586	1.00 × 10^−2^	2
*MCC*	5q22.2	45	3.79 × 10^−18^	1.41 × 10^−17^	1	rs6890833	3.40 × 10^−2^	2
*PRDM2*	1p36.21	31	4.55 × 10^−18^	1.41 × 10^−8^	1	rs2235515	2.00 × 10^−2^	2
*POR*	7q11.23	20	5.42 × 10^−21^	2.25 × 10^−13^	1	rs10262966	3.00 × 10^−2^	2
*KCNJ6*	21q22.13	17	8.09 × 10^−18^	8.25 × 10^−15^	1	rs4817896	2.40 × 10^−2^	2
*SORBS1*	10q24.1	17	8.00 × 10^−9^	5.09 × 10^−23^	1	rs10450393	1.00 × 10^−2^	2
*SHBG*	17p13.1	14	3.97 × 10^−12^	5.12 × 10^−5^	1	rs858524	3.00 × 10^−2^	2
*VDR*	12q13.11	14	1.39 × 10^−18^	6.75 × 10^−3^	1	rs731236	3.00 × 10^−2^	2

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
