# Peer review of "Deconvolution of the Genomic and Epigenomic Interaction Landscape of Triple-Negative Breast Cancer"

_cancers, 2019, doi:10.3390/cancers11111692_

Round 1

Reviewer 1 Report

The article entitled “Deconvolution of the genomic and epigenomic interaction landscape of triple negative breast cancer” by Wu et al described an integrative bioinformatics approach to understand the genomic and epigenomic alterations in triple negative breast cancer, using publicly available databases. The author’s have identified a complex array of genomic interactions and epigenomic changes that are associated with triple negative breast cancer. They also carried out network and pathway analysis to identify the functionally related genes, molecular networks, and potential significant pathways. The study is quite interesting, and the reviewer feels the manuscript can be accepted for the publication. The reviewer suggests including a table into the main manuscript, describing the discovery data sets utilized in the study. Including the number of cases/controls used from each data set; type of data and the platform used to generate the data; the criteria used to include in the analysis etc.

Author Response

We wish to thank the reviewer for the valuable comments as they have helped improve the quality of our manuscript.

Reviewer 2 Report

A significance of the study is "an urgent need for the discovery of molecular markers and targets for the development of novel targeted therapeutics."

A major strength of the study reported is the multiple hypotheses that are presented and tested. The authors provided data to support the hypotheses tested. 

The report includes multiple hypotheses that are tested which are classified as primary hypothesis and working hypothesis. 

Below are the hypothesis statements extracted from the manuscript.

1. We hypothesized that genes transcriptionally associated with TNBC are aberrantly methylated and that epigenomic alterations in these genes could potentially impact their expression.

2. We hypothesized that aberrant methylated genes transcriptionally associated with TNBC harbor somatic mutations.

3. Our primary hypothesis in this investigation was that TNBC is a complex disease influenced by both inherited variants in the germline DNA, somatic mutations acquired during tumorigenesis and epigenetic alterations.

4. Our working hypothesis was that significantly differentially expressed aberrantly methylated genes associated with TNBC harbor both germline and somatic mutations.

5. Our working hypothesis was that TNBC originates from a complex interplay between genomic (both germline and somatic mutations) variation and epigenomic variation, and that these complex arrays of interacting genomic and epigenomic factors affect entire molecular networks and biological pathways which in turn drive TNBC.

The manuscript could be improved if the hypothesis are clearly mapped to the sections in the methods and results. 

Hypothesis 5 above is the most comprehensive and demonstrates the integration of multiple data sources. 

Minor revisions

Line 40 to 41: "It affects primarily younger premenopausal women and tends to have higher incidences in African women,"

Should this sentence be African American women instead of African women based on the complete sentence?

Line 289: Table 1 should start on a new page.
Line 300: Figure 3 and Legend should be on same page.

Author Response

We are very thankful to the reviewer for the positive review, and excellent comments and suggestions that have helped us to greatly improve the quality of our manuscript.

Thank you 

Reviewer 3 Report

The authors’ goal was to dissect the overlap between germline mutation, cancer somatic mutation, CpG methylation, and altered transcription in TNBC.  Using TCGA and GWAS data, they sequentially filtered genes for enrichment in TNBC with regards to expression and methylation, then cancer somatic mutation, and finally germline mutation.  This was presented as a series of Venn diagrams and as lists of the most significantly altered genes in the various overlap categories.  The overlap categories reveal that tumor suppressors are often not only mutated, but also have altered methylation and expression.  Network analysis revealed interactions between genes with genomic alterations, methylation changes, or both, indicating the cancer cell phenotype emerges globally. This suggests that attempts to identify mechanisms of oncogenesis by somatic mutation alone, expression level alone, or methylation alone may be improved by more comprehensive methods as demonstrated here.  Alternatively, variance in a population that remains unexplained when using a single method may be better explained by more comprehensive methods.  A weakness is that the methylation coding does not discriminate between promoter hyper/hypo methylation and/or gene body hyper/hypo methylation.  It has been reported that promoter methylation tends to be silencing while gene body methylation tends to be up-regulating, and it may be informative to look at the classes of CpG individually if the bioinformatics tools allow it, or to further clarify the characteristics of the CpGs evaluated.   Overall, this deconvolution offers valuable insight into the landscape genomic and epi-genomic alterations, and I would like to see the output of similar analyses performed on other cancers. 

This type of deconvolution appears to be novel in the context of TNBC. The introduction provides relevant background for the context of the research, with accompanying citations. The writing is very good, needing only a few minor proofreading edits. As the authors are primarily offering a methodological tool, I do not think that they are overstating any conclusions. Further research will tell whether the assumptions underlying their modeling can be improved upon, but for now their diagrams as presented give an interesting visual breakdown of the types of genome/methylome/expression alterations occurring on TNBC.   In-text description of Fig 2B needs to be checked, i.e. text states “162 genes were hyper methylated and 408 genes were hyper methylated”. Authors’ likely intent was “162 genes were hyper methylated and down-regulated and 408 genes were hyper methylated”. The legend on Fig 2B indicates that a single color was used for 2 quadrants: “Hypo-methyl, Down-regulated + Hyper-methyl, Up-regulated (81)”. The color coding is fine because the quadrant layout is easy to determine, but please make it clear whether the “(81)” refers to the sum of both categories, or to only the Hyper-methyl, Up-regulated group.  As it is presented, latter interpretation seems more likely to be the intent, but it is not entirely clear.  The same concern applies for Fig 2B: “Hypo-methyl, Up-regulated + Hyper-methyl, Down-regulated (162)”. In-text description of Fig 4 needs to be checked against the figure. Some values and categories appear to be mismatched in the text.

Author Response

We appreciate the reviewers time and helpful comments.  

They have helped us significantly improve the quality of the manuscript.

Thank you.

Reviewer 4 Report

In this study, the authors integrate a variety of GWAS studies with several types of TCGA data for 83 TNBC samples and 83 normal breast samples.

There are a number of issues with this manuscript.

1) The numbers shown in Figure 2 for the various categories of DM and DE genes do not match up.  Something is missing here.

2) It does not appear that the authors have selected significantly recurring somatic mutations, but have taken any somatic mutations. There is no correction for hypermutator cases with loss of MSH2 or other DNA repair proteins.  There doesn't seem to be a correction for the size of genes, since TTN (the largest gene in the genome without any known role in cancer) appears at the top of the list in Table 2.

3) The authors suggest that the genes identified could be biomarkers, but there are no analyses to associate them with outcome.

4) There are no analyses of somatic or germline variations to determine if they serve as eQTL's or mQTL's.  See PMID:28096648.

5) There is no adjustment for where in the gene the CpG's are found in the DM analyses.  Does it matter if they are in the islands, shores, or body of the genes? How does that affect the analyses in Figure 2?

6) There is no validation in other cohorts (e.g. METABRIC).

7) There is explanation of how this study is novel, given several earlier integrative studies on TCGA Breast Cancer datasets.  See PMID:29622464, 30866861, 27690302, 27070496, and 28096648.

Minor comments:  There are a number of grammatical and typographical errors throughout the manuscript.

Author Response

We very much appreciate the time and helpful comments and suggestions of the reviewer's comments and all the insights that helped us to think very carefully and improved the manuscript including the discussion.  

Round 2

Reviewer 4 Report

The authors have not addressed the critiques from the initial review. 

Specifically, they have not addressed hypermutator phenotypes, gene size, recurrent mutations, eQTL's, mQTL's, or lack of novelty.